# Teaching–Learning-Based Optimization Algorithm Applied in Electronic Engineering: A Survey

Kenia Yadira Gómez Díaz , Susana Estefany De León Aldaco *, Jesus Aguayo Alquicira, Mario Ponce-Silva and Víctor Hugo Olivares Peregrino

Centro Nacional de Investigación y Desarrollo Tecnológico (CENIDET), Cuernavaca 62490, Mexico
* Correspondence: susana.da@cenidet.tecnm.mx

**Abstract:** This paper presents a survey of the articles published in the period 2013–2021 related to the application of the teaching–learning-based optimization (TLBO), which reproduces the dynamics that occur in a classroom with the teacher and the student. This paper uses the algorithm to optimize some objective functions related to the design in the electronics field. A total of 62 papers were reviewed and some graphs were generated to summarize the most relevant of these articles. These have been classified into five categories based on the areas of electronic engineering, such as power electronics, control, electromagnetism, digital electronics, and analogue electronics. Electronic engineering has been becoming increasingly relevant in world technological development, for example, in electric vehicles or generating electricity from renewable energy sources to counteract the environmental impact that non-renewable sources generate. The TLBO algorithm has attracted the interest of a large number of researchers due to its efficiency, speed, and low initialization parameter requirements. This article is composed of two stages, the first is a summary of the information on electronics, in general, encompassing all its areas, and the second focuses on the algorithm applied to multilevel inverters; for each stage, graphs and tables are shown.

**Keywords:** bio-inspired computing; multilevel inverter; optimization; TLBO algorithm; power converter; power electronics

## 1. Introduction

Optimization can be defined as the search for the solution to a problem in which it is necessary to maximize or minimize a single-objective function (single-objective) or a set of them (multi-objective) within a domain containing the values of acceptable variables (decision variables), while some constraints must be satisfied [1]. Its main task is to find the best possible solution to a specific problem. Optimization of a product or process is the determination of the conditions that result in its optimal performance, based on the optimization parameters introduced in the mathematical formulation of the actual model [2]. Optimization can be implemented in many areas, such as engineering, design, control, and economics, among others. However, the fact that it can be optimized in all these areas does not imply that it is an easy task to solve, as some engineering problems are very complicated.

In 1939, the first linear programming algorithm was developed and used in the area of economics by L. Kantorovich, who formulated the optimal planning production problem and efficient methods for finding solutions using linear programming. For this work, he shared the noble prize with T. Koopmans in 1975 [3].

To optimize, an objective function is required, which is the mathematical equation that describes the problem; it must also be defined whether it is intended to minimize or maximize the problem, the decision variables, and their respective constraints, based on which parameters or limits to the algorithm will make decisions. It should be added that the algorithms must be programmed using a programming language.

When dealing with mathematical equations that represent the problem to be optimized, some methods and algorithms can be classified into three types [4]:

1.  Algebraic methods: such as force summation and symmetric polynomial theory, among others.
2.  Numerical methods: such as Newton–Raphson, gradient optimization, and homotopy algorithm, among others.
3.  Metaheuristic algorithms: such as genetic algorithm (GA), differential evolution (DE), particle swarm optimization (PSO), and teaching–learning-based optimization (TLBO) [5], among others.

The algebraic method and the numerical method are included in a single category, called classical or conventional methods. Metaheuristic or bio-inspired algorithms, on the other hand, are more recent, but their implementation has been increasing in some industrial areas. The main difference between classical and metaheuristic methods is the speed with which they solve non-transcendental linear equations because they are faster and more effective in optimizing engineering problems.

There are several ways to classify metaheuristic algorithms. Depending on the characteristics selected to differentiate them, several classifications are possible, and the result will be from a specific point of view. Classification into nature-inspired versus nature-inspired metaheuristics, memory-based versus memoryless methods, or methods using a dynamic or static objective function is possible. In this overview, this will be done according to the single-point versus population-based search classification, which divides metaheuristics into trajectory methods and population-based methods, allowing for a clearer description of the algorithms.

The term trajectory method is used because the search process performed by these methods is characterized by a trajectory in the search space, the performance of which is usually rather unsatisfactory. They incorporate techniques that allow the algorithm to escape local minima. This implies the need for termination criteria other than simply reaching a local minimum.

Population-based methods deal with a set or population of solutions at each iteration of the algorithm rather than a single solution. In this way, population-based algorithms provide a natural and intrinsic way of exploring the search space. However, the final performance is highly dependent on how the population is manipulated [6].

Figure 1 shows a classification of optimization methods, the one this research will focus on is the teaching–learning-based optimization (TLBO) algorithm, which is found in the population-based, nature-inspired metaheuristic and non-reproductive types.

The objective of this paper is achieved by conducting a literature survey. According to the Association for Computing Machinery, a survey is "A paper that summarizes and organizes recent research results in a novel way that integrates and adds understanding to work in the field. A survey article assumes a general knowledge of the area; it emphasizes the classification of the existing literature, developing a perspective on the area, and evaluating trends" [7]. In other words, it develops a perspective of the area but does not go into depth and analyze each of the articles, as is the case in systematic reviews or reviews of the state of the art.

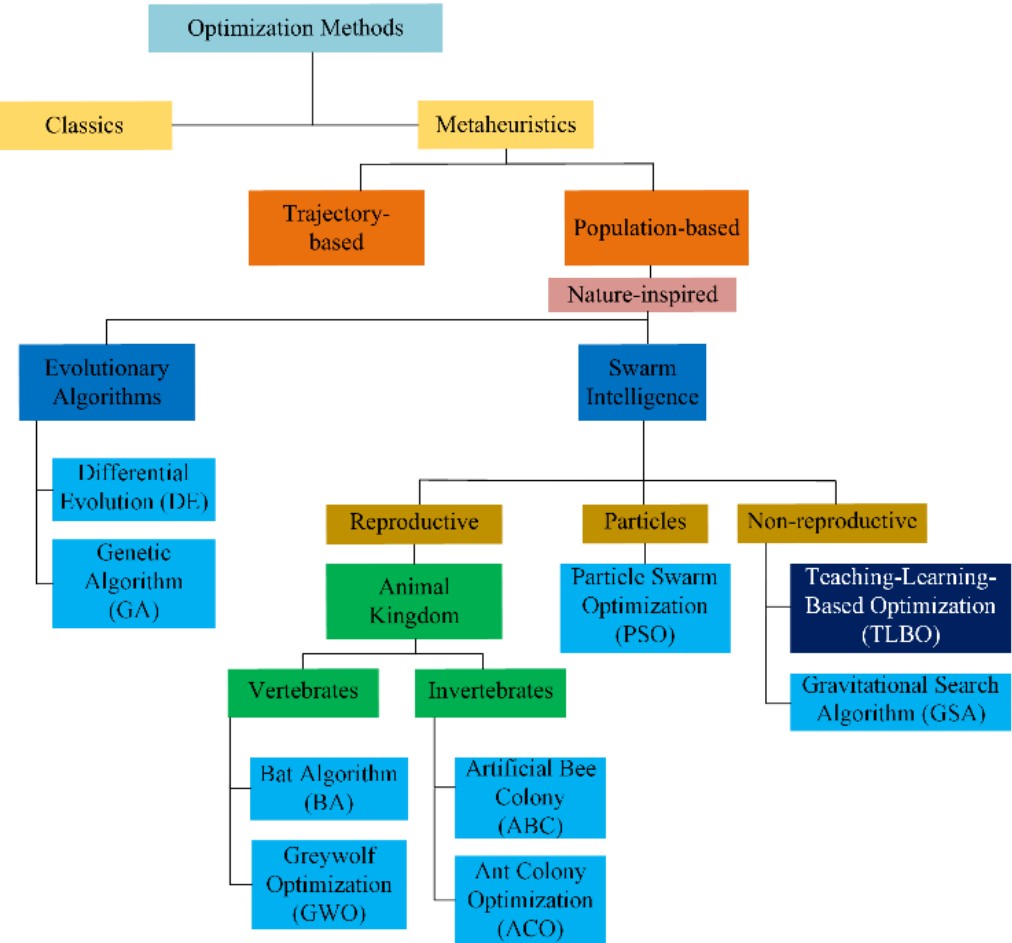

**Figure 1.** Classification of optimization methods.

This paper focuses on providing a survey of recent optimization works and applications of the TLBO optimization method in the area of electrical engineering, as opposed to previous reviews that exist in the literature that summarize all areas in general. The main contributions of this paper are as follows:

- The paper aims to provide a survey on the recent progress and application of the TLBO algorithm in the area of electronics. This is rarely found in previous works, allowing beginners to become familiar with the TLBO algorithm.
- The article provides a taxonomy of the TLBO algorithm, which is useful for readers to understand and apply the TLBO algorithm.
- The article describes the fields of application and the solutions obtained by the TLBO algorithm. All these are useful for understanding the algorithm and are expected to benefit both practical applications and future research.

## 2. Methodology

As a starting point, a search in various databases for surveys, systematic reviews, and state of the art reviews focused on the use of the TLBO algorithm in different areas was carried out. The articles found are classified in Table 1.

**Table 1.** Reviews of the TLBO algorithm.

| Reference | Year | Article | Area |
|---|---|---|---|
| [8] | 2019 | A survey on teaching–learning-based optimization algorithm: short journey from 2011 to 2017 | Electrical engineering, data mining, optimization, and other applications |
| [9] | 2015 | A Short Survey on Teaching Learning Based Optimization | Optimization method for continuous non-linear large-scale problems constrained and unconstrained real parameter optimization problems shape and size optimization of truss structures with dynamic frequency constraints |
| [10] | 2018 | A survey of teaching–learning-based optimization | Manufacturing and operation research, mechanical and electrical engineering, civil engineering, electronics and control engineering, pattern recognition and image processing, other areas |
| [11] | 2019 | A Survey of Application and Classification on Teaching-Learning-Based Optimization Algorithm | General: Shop Scheduling, power systems, truss structures, multi-objective optimal, two-sided assembly line, others |
| [12] | 2015 | An improved teaching-learning-based optimization: briefly survey | Economic Load Dispatch Problems |
| [13] | 2012 | Population based meta-heuristic techniques for solvingoptimization problems: A selective survey | Location of automatic voltage regulators in distributed systems, integer programming for generation maintenance scheduling in power systems, data clustering, economic load dispatch problema with incommensurable objectives |
| [14] | 2019 | A survey on new generation metaheuristic algorithms | Data classification, quadratic assignment problems, design procedure, size and shape of structures |
| [15] | 2016 | Review of applications of TLBO algorithm and a tutorial for beginners to solve the unconstrained and constrained optimization problems | Encompasses many areas |
| [16] | 2017 | Applications of TLBO algorithm on various manufacturingprocesses: A Review | Machining processes: ultrasonic, electro chemical, electrical discharge, aser beam, electron beam, water jet, abrasive jet |
| [17] | 2017 | Review of the Teaching Learning Based Optimization Algorithm | Term hydrothermal scheduling problema, dynamic economic dispatch, flow shop and job shop scheduling, and others |

As a result of this first classification shown in Table 1, the idea of developing a survey article on the applications of the TLBO algorithm in the field of electronic engineering arose because it was identified that there was a lack of such articles.

This article presents a synthesis of some applications of the teaching–learning-based optimization (TLBO) algorithm to the area of electronic engineering, analyzing, and classifying publications from the period between 2013–2021.

The search for publications was initiated in various databases and search engines, such as IEEExplore, Springer Nature, ScienceDirect, and SciELO, among others. The universe of publications analyzed and classified is 62 papers related to optimization using the TLBO algorithm in the area of electronic engineering.

The selection process of the articles to be reviewed is shown in the flow chart of Figure 2, where you can understand the steps followed for the selection of the 62 articles, one of the first steps was the selection of the keywords, which were "TLBO electronic engineering", "TLBO Multilevel Inverter", "Electronic TLBO", "Electronic optimization TLBO".

In Table 2, the areas, years, references, and applications that can be found in the literature on the TLBO algorithm in electronic engineering are shown; in the second area of this article there will be more information about all the papers and some examples of the problem application that some authors optimize with this algorithm.

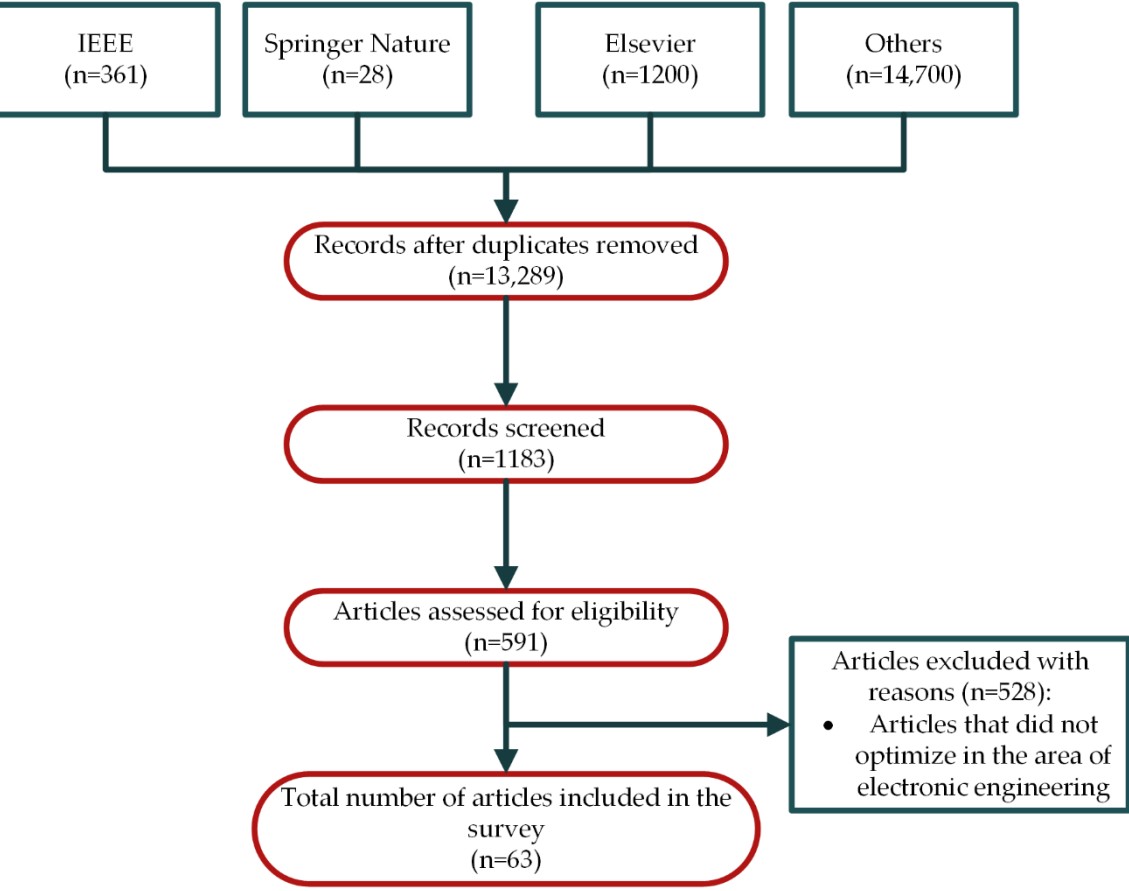

**Figure 2.** Flow chart of the selection of the articles to review.

**Table 2.** The TLBO algorithm used in the area of electronic engineering.

| Area | References | Applications | Years |
|---|---|---|---|
| Power Electronics | [18–56] | Power generation and distribution, multilevel inverters, other | 2014–2021 |
| Control | [57–68] | PID, other | 2014–2020 |
| Electromagnetism | [69–72] | Electrical machines and other | 2016–2019 |
| Digital Electronics | [73–75] | Filters and cameras | 2013–2020 |
| Analog Electronics | [76–78] | Antenna and filters | 2016–2020 |

## 3. Results

### 3.1. The Teaching–Learning-Based Optimization Algorithm

The TLBO algorithm was originally introduced by V. Rao in 2011 and is inspired by the philosophy of the teaching–learning process in a classroom and imitates the influence of a teacher on student outcomes. Like other swarm intelligence algorithms, TLBO is a population-based metaheuristic optimization algorithm. It has been very popular due to some characteristics of the TLBO algorithm, such as its concept and that it does not require specific parameters, it is fast and easy to implement, and it has been widely applied to solve numerous problems in various engineering areas [10].

The TLBO algorithm is based on the effect of a teacher's influence on the performance of students in a class. The algorithm describes two basic methods of learning the first through the teacher (known as the teacher phase) and the second through interaction with

other learners (known as the learner phase). In this optimization algorithm, a group of learners is considered a population and the different topics offered to the learners are considered as different design variables of the optimization problem and the outcome of a learner is analogous to the "fitness" value of the optimization problem. The best solution for the whole population is considered to be the teacher. The design variables are the parameters involved in the objective function of the given optimization problem and the best solution is the best value of the objective function [1]. In Figure 3, the flowchart of the TLBO algorithm is shown. Results may be divided into subheadings. It should provide a concise and precise description of the experimental results, their interpretation, as well as the experimental conclusions that can be drawn.

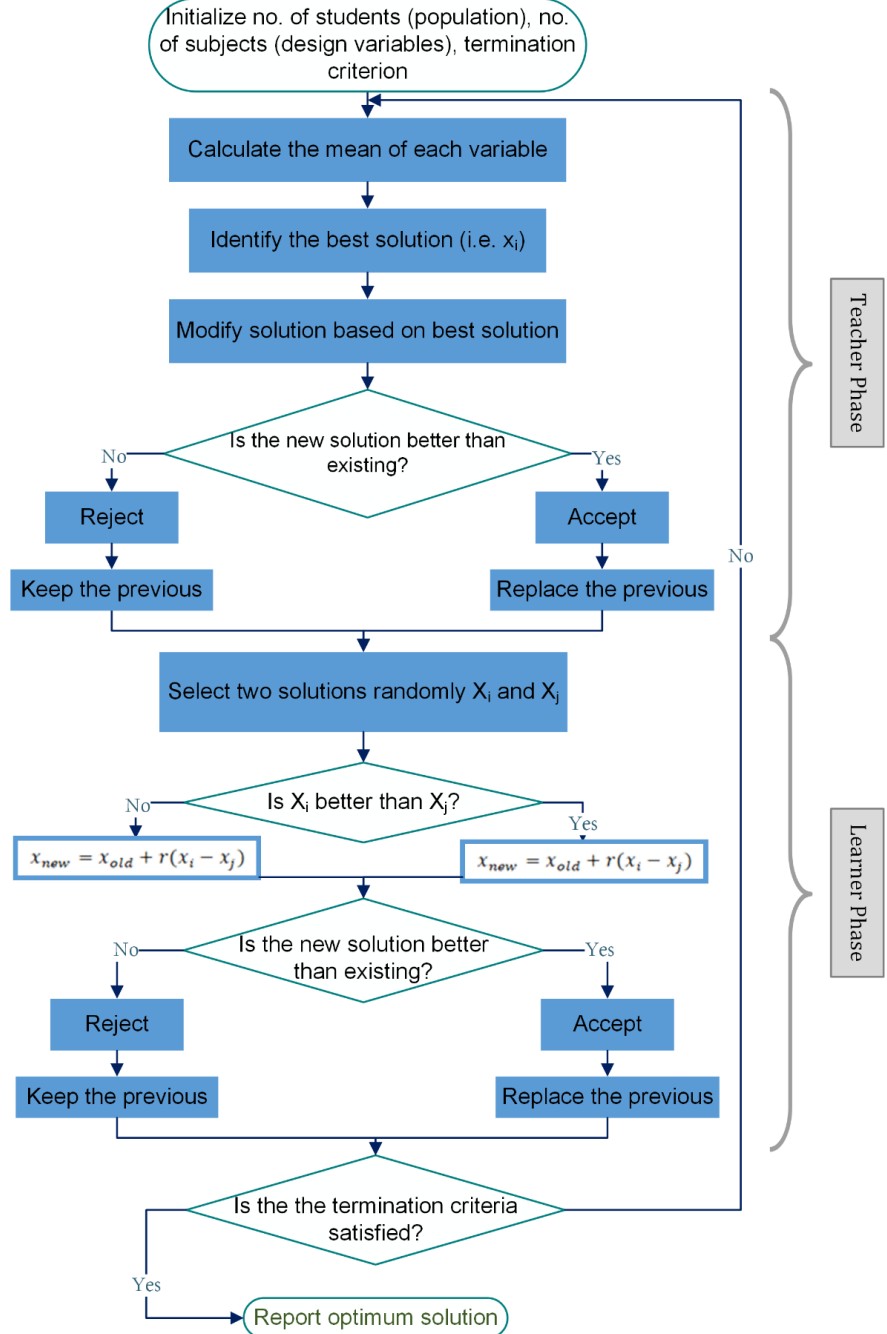

**Figure 3.** Flowchart of the TLBO algorithm [18].

1.  Initialization: the user provides a population (number of students), the decision variables or design variables (number of topics), and the termination criteria is the maximum number of iterations.
2.  Teacher phase: This is the first part of the algorithm in which the students learn through the teacher. During this phase, the teacher tries to increase the average class score in the subject he/she teaches according to his/her ability. In any iteration i, assume there are "m" number of subjects (i.e., design variables), "*n*" number of students (i.e., population size, k = 1, 2, ..., *n*), and $M_{j,i}$ is the average result of the students in a particular subject "j" (j = 1, 2, ..., m). The best overall result $X_{total-kbest,i}$ considering all subjects together, obtained in the whole population of students, can be considered as the result of the best student kbest. However, as the teacher is usually considered to be a highly educated person who trains the students so that they can obtain better results, the algorithm considers that the best student identified is the teacher.
3.  Learner phase: This is the second part of the algorithm in which learners increase their knowledge by interacting with each other. A learner interacts randomly with other learners to improve his or her knowledge. A learner learns new things if the other learner has more knowledge than him/her. The one that provides the best result to the function is the one that will be chosen and will end the process when the termination criterion that was set in the initialization stage is met.

All the previously mentioned information allows us to say that the TLBO algorithm is developed in the following steps:

(a) Formulation of objective function or fitness function.
(b) Initialization of optimization parameters and the limits of the variables.

Generation of a random population. The population is expressed as:

$$Population = \begin{bmatrix} x_{11} & x_{12} & \cdots & x_{1D} \\ x_{21} & x_{22} & \cdots & x_{2D} \\ \cdots & \cdots & \cdots & \cdots \\ x_{P_n,1} & x_{P_n,2} & \cdots & x_{P_n,D} \end{bmatrix} \tag{1}$$

(c) Teacher phase: the mean for the particular variable can be calculated as the following equation:

$$M_{*,D} = [m_1, m_2, \ldots\ldots, m_D] \tag{2}$$

(d) The best solution will be considered as a teacher for that iteration:

$$X_{teacher} = X_{(f(X)=\min)} \tag{3}$$

(e) Sort the grade point of each variable of each student and a new wean is calculated. The difference between two means can be calculated with next equation, $T_F$ may be considered as 1 or 2.

$$Difference_{*,D} = r(M_{new,D} - T_F \times M_{*,D}) \tag{4}$$

(f) Update the values by adding the difference to the old solution.

$$X_{new,D} = X_{old,D} + Difference_{*,D} \tag{5}$$

(g) Learner's phase, in this second phase, the knowledge transfer takes place between the mutual interactions between the learners. The mathematical equations are as follows:

$$\begin{cases} X_{new,i} = X_{old,i} + r_i\left(X_i - X_j\right) \\ X_{new,i} = X_{old,i} + r_i\left(X_i - X_j\right) \end{cases} \tag{6}$$

(h)　The process will be finished only if the maximum generation is reached, otherwise repeat all the process.

### 3.2. *The Teaching–Learning-Based Optimization Algorithm Applied in Electronic Engineering*

Figure 4 shows the percentage distribution of the classified publications according to whether they are journal or conference publications. According to the figure, 77% are journal publications.

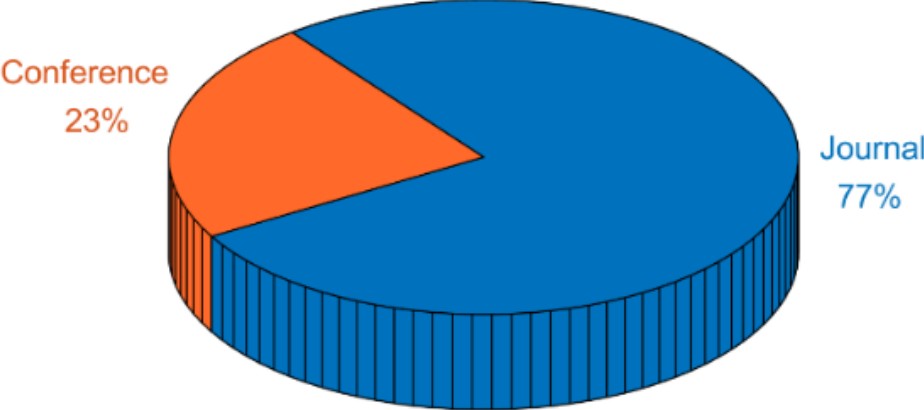

**Figure 4.** The graph represents the percentage of papers in journals and conferences.

This research focuses on the state of the art of the TLBO algorithm applied in sub-areas of electronics, such as power electronics, electrical or electronic control, analogue electronics, digital electronics, and electromagnetism.

Figure 5 shows a histogram with the number of publications per year of publication in the publication of TLBO algorithm articles.

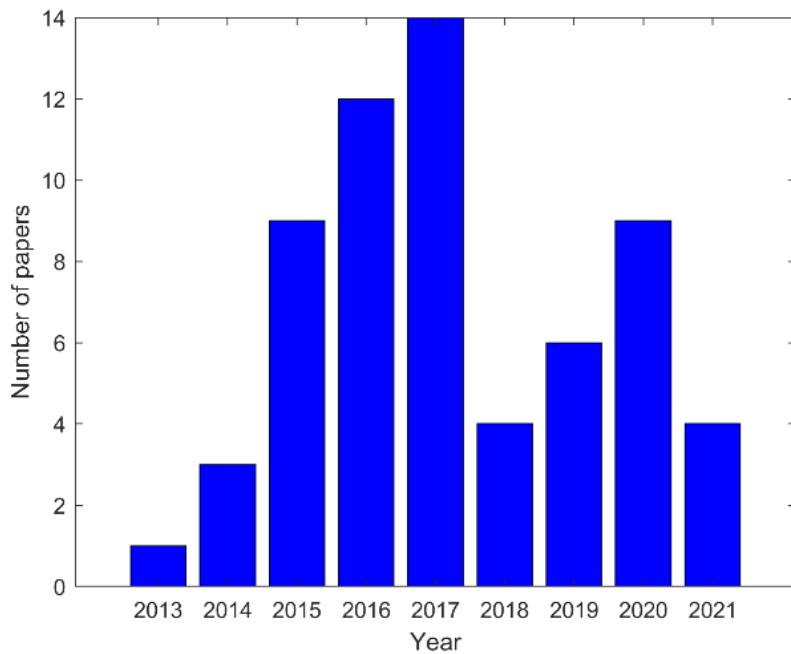

**Figure 5.** Graph of the years of publication of the articles implementing the TLBO algorithm.

The search for publications in the various databases focused on the application of the TLBO algorithm in the area of electronic engineering. To organize, the information collected was divided into five categories: power electronics, control, electromagnetism, digital electronics, and analogue electronics. Figure 6 shows the percentage distribution for each category.

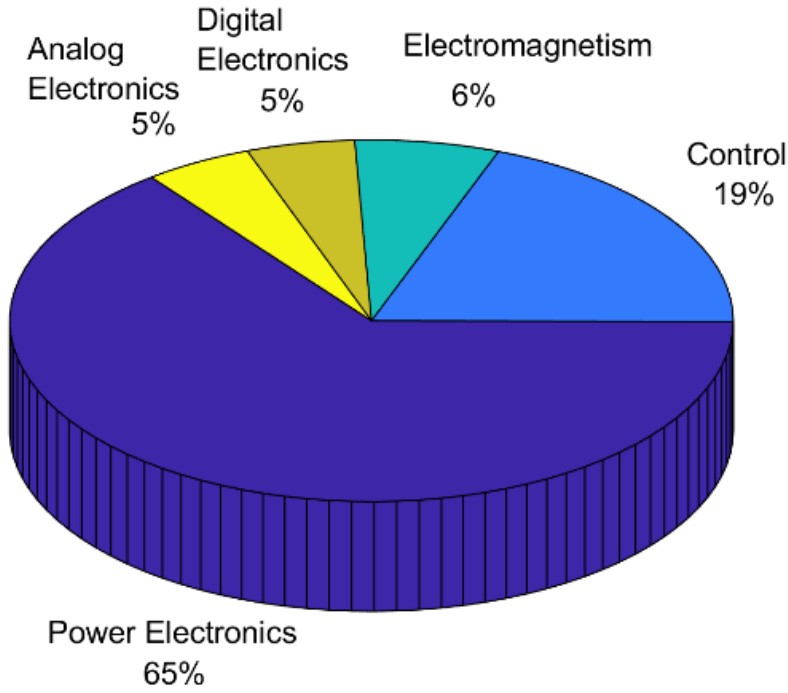

**Figure 6.** The graph represents the percentage of the areas that the papers belong to.

According to the distribution shown in Figure 6, 65% of the publications belong to the power electronics category. Each of the areas will be dealt with in different sections in which a table will be found with a description of what has been done in each of the articles and a summary of what each area encompasses.

*3.3. The TLBO Algorithm in Power Electronics*

Power electronics is the processing, control, and conversion of electrical energy through the use of semiconductor devices that operate as switches.

Statistics were generated for the power electronics category divided by the type of application into multilevel inverters, power generation and distribution, and others. The realization of the tables was divided into two; Table 3 is about power generation and distribution and others, Table 4 is about multilevel inverters.

Figure 7 shows the percentage distribution of the power electronics category, it is divided into three subareas that are multilevel inverters, power generation and distribution, and others.

The largest contribution with 45% corresponds to the type of applications related to energy generation and distribution, which includes STATCOMs, distribution networks, energy system design, and maximum power point monitoring, among others. Thirty-five per cent corresponds exclusively to multilevel inverters. Finally, the contribution of 20% corresponds to applications labelled as "others", where topics, such as obtaining parameters of photovoltaic models, hybrid AC/DC power systems, DC–DC converters, electric vehicles, and power factor compensation, are addressed.

It is of particular interest to analyze in more detail the articles grouped in the multilevel inverter application type because the 14 articles that make up 35% of the subareas of power electronics belong to the same application topic, compared to the area of power generation

and distribution, which despite being the majority, involves more application topics, such as distribution networks, power system design, and global maximum power point tracking.

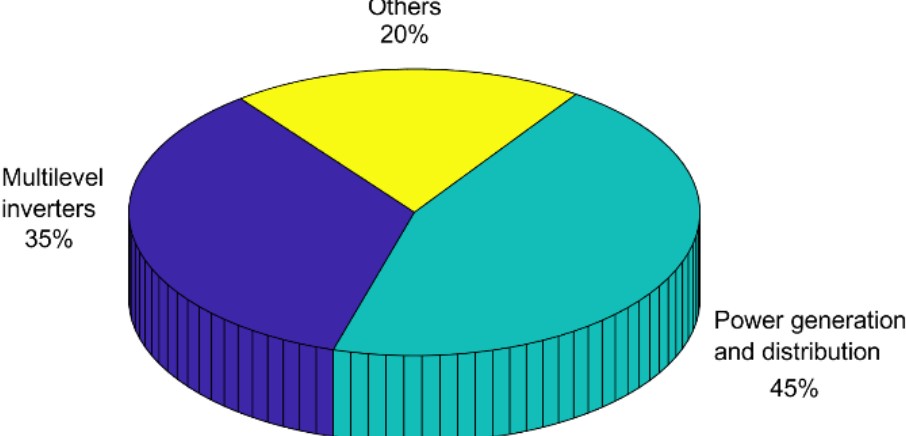

**Figure 7.** The graph represents the power electronics subareas.

**Table 3.** The TLBO algorithm used in power electronics.

| Area | Ref | Year | Application | Description | Author's Conclusion |
|---|---|---|---|---|---|
| | [19] | 2020 | DAB converter | TLBO used to design the system with a minimum volume inverter and achieve maximum power efficiency | TLBO has better results and a faster convergence rate |
| | [20] | 2019 | PFC and cloud data logger monitoring | TLBO used to optimize the capacitor bank and compensate power factor | TLBO reduced the time to find the best solution and obtained the power factor near the desired level |
| | [21] | 2017 | Load dispatch with TLBO | TLBO used to reduce energy production costs in power plants through economical load dispatching | TLBO provides lower-cost and high-quality solutions to the problem of economical freight forwarding |
| | [22] | 2015 | Design of a hybrid energy system | TLBO is used to design a hybrid system that can supply a typical household load. | TLBO was able to find the optimal design parameters and the system can provide power at an acceptable cost |
| | [23] | 2016 | Distribution network | Optimize system performance, such as real power loss, voltage profile, and voltage stability index | TLBO has shown good performance and efficiency, compared to the PSO and GA method |
| Power Electronics | [24] | 2016 | A DSTATCOM | Optimize the PI controller gains and filter parameters and compare the efficiency of the algorithms (TLBO and GEM) | TLBO performs better and converges faster than the GEM. The current THD is below the IEEE 519 standard. |
| | [25] | 2016 | Maximum power point global tracking (MPPT) | Improve TLBO algorithm for use in PV modules, compare results with other algorithms | I-TLBO allows to automatically adjust the teacher's values, but it takes longer than the TLBO, but it does increase its power by 0.6 W |
| | [26] | 2019 | Dynamic voltage recloser (DVR) design | TLBO for the design to select the optimum nominal voltage values of the DVRs used in an IDVR to compensate for balanced voltage drops | Both methods lead to the optimal solution. However, the accuracy of TLBO is higher than GA and in less time |
| | [27] | 2017 | Flexible alternating current transmission systems "STATCOM" | Minimize transmission losses, determine the optimum value of control variables (real and reactive power) | TLBO has the efficiency to reduce the active power loss reasonably without violating any restrictions. It also has excellent convergence characteristics |
| | [28] | 2016 | Maximum power point trackers (MPPT) | The TLBO algorithm seeks to optimize the output power of the PV array as a function of the duty cycle of the DC-DC converter | The TLBO algorithm converges slower than the MBA, so it is less efficient |
| | [29] | 2019 | Parameter extraction from photovoltaic models | Extract the parameters of photovoltaic models using ITLBO and compare their performance with TLBO and others | ITLBO can provide more accurate and reliable parameter values, but the tables show that in some categories it has the same result as TLBO |

**Table 3.** *Cont.*

| Area | Ref | Year | Application | Description | Author's Conclusion |
|------|-----|------|-------------|-------------|---------------------|
| | [30] | 2017 | Modified energy-efficient localization | Application of TLBO in MDV-Hop, Calculation and modification of the hop size, Optimal selection of anchor nodes, Updating the location | TLBO achieves higher localization accuracy. The results show that it has better positioning coverage and high location accuracy with lower power consumption |
| | [31] | 2021 | AC/DC hybrid power system | The evaluation function examines the suitability of the degree point as a solution to the optimization problem | It is tested on IEEE14, 30, and 57 bus systems, and it is observed in the tables that TLBO is the reduction in TFC, TRPL, and HVSI |
| | [32] | 2017 | Operational scheduling of renewable energy sources | TLBO aims to optimize the search for the lowest total cost of operation for 24 h | The results showed that TLBO is an effective method and achieves a lower cost (7–11%) than the other methods |
| | [33] | 2017 | Operation of the pumped-storage hydroelectric power plant | TLBO seeks to minimize the operating cost of the hydroelectric power plant | TLBO obtains the optimal generation considering the wind farm and the PHS unit |
| | [34] | 2017 | MPPT technology in the solar photovoltaic system | TLBO aims to obtain at any time the maximum power from a photovoltaic panel and to design an efficient DC–DC converter | The proposed TLBO-MPPT method gives more efficient results in the designed DC–DC converter, compared to conventional techniques |
| | [35] | 2017 | Hydrothermal energy system | TLBO aims to determine the optimal power generation from hydro and thermal power plants to minimize the total operating cost of thermal power plants | The results have been obtained for water discharge, reservoir storage volume, and optimal MW values of real hydroelectric and thermal power |
| | [36] | 2016 | Reconfiguration of the primary power distribution system | Aims to develop a TLBO-based bit relocation controller for simultaneous reconfiguration of primary PDSs | The best-reconfigured system has an active power loss of 0.130 MW, which is reduced by 30.10%, with respect to the base network |
| | [37] | 2018 | Real power generation for congestion | TLBO aims to minimize congestion cost while satisfying the network constraint | Total congestion cost (USD/h) = 494.66 |
| | [38] | 2015 | DC-DC converter on the RFID tag | TLBO aims to increase the voltage conversion ratio and also reduce the size of the charge transfer capacitor | The output voltage and voltage conversion ratio of the 4-stage DC–DC converter are 7.741 V and 86.01% |
| | [39] | 2017 | Electrical systems troubleshooting | TLBO aims to estimate the location of faults in electrical systems and also to obtain the error rate | TLBO achieves almost the same accuracy in a very short time, which is 7 s and can detect and locate any type of fault |
| | [40] | 2015 | Small-signal stability of a wind system with DFIG | TLBO aims to formulate state space model of DFIG, investigation of fault passage capability | Using TLBO is much better than PSO for the power system model |
| | [41] | 2016 | Simulation of global MPPT for photovoltaic systems | TLBO aims to track the global MPP (maximum power points) of PV power system under partial shading condition (PSC) | Simulations performed under varying shading patterns reveal that the performance of the TLBO-based tracker is better than PSO and fuzzy logic control (FLC) |
| | [42] | 2016 | Reactive power planning | TLBO is applied to determine the size of reactive power sources | From the results obtained, it is found that the active power loss is minimal in the TLBO method |
| | [43] | 2017 | Microgrid energy management | TLBO aims to develop an energy management infrastructure for sources and loads such that the total cost required is reduced | TLBO can achieve very accurate results in virtually any circumstance. Compared to others, it is the best solution finder. |

In the multilevel inverter articles, the three most commonly used multilevel inverter (MLI) topologies are presented, which are latching diodes, floating capacitors, and cascaded H-bridge. In which they sought to reduce the percentage of total harmonic distortion (THD). The main reason for reducing the THD in the output signal of an MLI is the problems that occur when a high THD percentage is obtained. Some of these problems are as follows:

1. Overheating of the drivers.
2. Malfunctioning electrical and electronic equipment.
3. Overheating in motors.

The effect of harmonics and unbalances in the system on the motors is mainly in the heating of the motor, causing losses in the core. In addition, it causes parasitic torques in the motor shaft, causing pulsating torques, i.e., vibration. Ultimately, this is a stress on the motor and leads to a reduction in the service life. On the other hand, in the case of electronic equipment, harmonic currents distort the voltages at the power supply nodes.

This voltage distortion causes the malfunction of more sensitive electronic devices, such as programmable logic controllers (PLC), control and process equipment where their synchronization depends on zero crossings of the voltage. Much of this equipment requires a completely clean power supply for proper operation [79].

Table 4 shows more detailed information from the articles related to multilevel inverters. Most of the articles compare the TLBO algorithm with other optimization algorithms, such as the marine predators algorithm (MPA); the flower pollination algorithm (FPA); a hybrid algorithm that combines the PSO algorithm with the grey wolf optimizer (GWO), which is summarized as PSOGWO; elitist teaching–learning-based optimization (ETLBO); PI control techniques coupled with the teaching–learning-based optimization (TLBOPI) algorithm; adaptive neuro-fuzzy inference system (ANFIS); fuzzy logic control (FLC); as well as classical methods, such as Newton–Raphson (NR) are also used. The *n*-level that is in the first column that represents the number of levels, and in the last column Ref means reference, CHMLI means cascaded H-bridge multilevel inverter, and DCMLI means diode-clamped multilevel inverter. It also shows the efficiency of the TLBO algorithm in this type of application can be observed, since in almost all the articles a THD percentage of less than 10% was obtained.

**Table 4.** The TLBO algorithm used in multilevel inverters.

| *n*-Level | Year | Load | Results (%THD) | Ref |
|-----------|------|------|----------------|-----|
| | | | CHMLI | |
| 5 | 2019 | R | Not specified | [44] |
| | 2017 | R | Line = 11.53 Phase = 10.33 | [18] |
| | 2018 | R | Case1 = 8.20 Case2 = 7.70 | [45] |
| | 2016 | R | NR = 8.86 TLBO = 6.95 | [46] |
| 7 | 2020 | R | FLC = 7.77 TLBO = 2.13 ANFIS = 1.68 | [47] |
| | 2015 | R | Line = 9.45 Phase = 13.3 | [48] |
| | 2017 | R | Line = 5.98 Phase= 18.89 | [49] |
| | 2021 | R | TLBO = 8.2 MPA = 5.5 FPA = 6.1 PSOGWO = 8.2 | [50] |
| | 2020 | R | TLBO = 5.2 PSO = 6.22 | [51] |
| | 2017 | R | TLBO = 5.95 | [52] |
| 27 | 2020 | R | ETLBO = 4.0 NR = 3.02 | [53] |
| | | | DCMLI | |
| 5 | 2016 | RL | 41.13 | [54] |
| 7 | 2014 | RL | TLBO = 8.1 ETLBO = 9.0 NR = 14.3 | [55] |
| | | | Modular | |
| 9 | 2020 | R | 7.3 | [56] |

In these articles, very relevant information was obtained about the number of levels, the type of load most used and how much the THD percentage was reduced with respect to other algorithms, some of these articles were obtained thanks to the optimization of the following objective function [18,49,52]:

$$10 \times |V_1^* - V_1| + \text{THD}_{Phase} \tag{7}$$

where $V_1^*$ is the modulation index of the fundamental component and varies from zero to one, $V_1$ is the modulation index that was used for the elimination of one of the harmonics, and $|V_1^* - V_1|$ is the absolute value of the error, required to adjust the fundamental harmonic. A weighting factor equal to ten was applied to the error terms to increase the importance of the fundamental component. The fundamental component almost reaches

the desired values, as well as the minimum possible THD at the output with the presented weighting factor. The decision variables would be the switching angles, which must meet certain characteristics presented in Equation (2); these can also be defined as constraints [18,49,52].

$$0 \le \alpha_1 \le \alpha_2 \le \alpha_3 \le \frac{\pi}{2} \tag{8}$$

Figure 8 shows the number of voltage levels with their respective percentage. As can be seen, the 7-level multilevel inverter is the most common, which has three decision variables, i.e., three switching angles; the more decision variables the greater the complexity of the problem to be optimized, which is why there is a lower percentage in the 9-level and 27-level inverters.

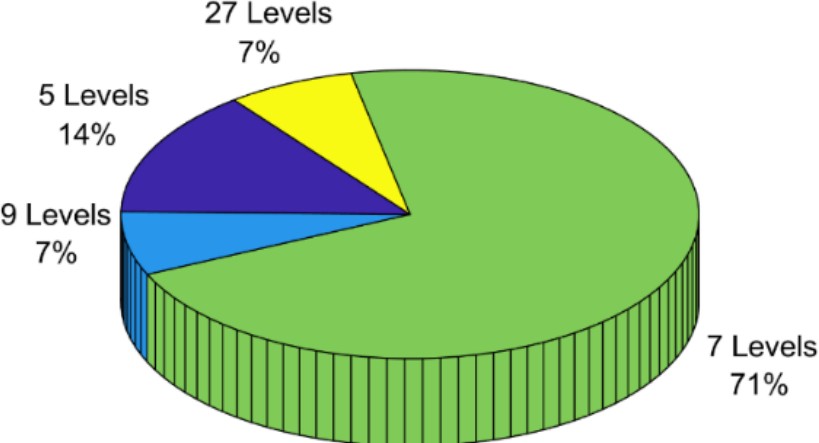

**Figure 8.** The graph represents the number of levels most commonly used by multilevel investors.

Figure 9 shows the percentage distribution according to the type of load used in the inverter. A total of 86% of the publications analyzed use resistive load (R), and 14% are for the resistive-inductive load (RL).

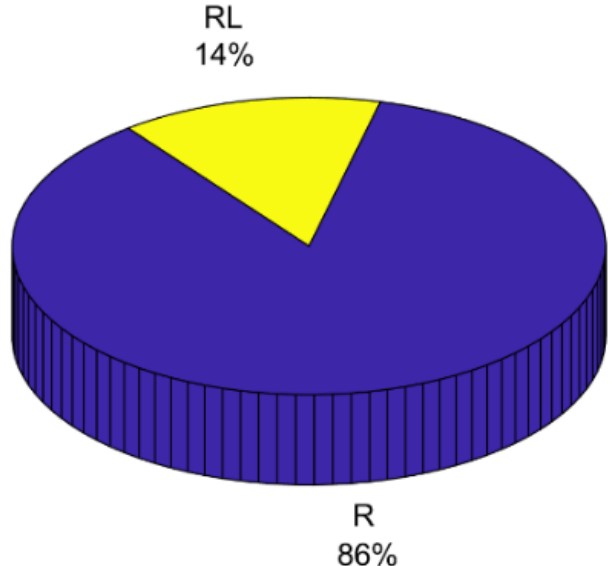

**Figure 9.** The graph represents the type of load most commonly used in multilevel inverters.

It is observed that there are recent articles on this type of application because the algorithm could be considered modern, which increases the interest to implement and study it in a wide variety of application areas [11], one example of that is in multilevel inverters, thanks to the characteristics that the algorithm has.

### 3.4. The TLBO Algorithm in Control

The control area was divided into three application sub-areas, PID, electrical systems, and others. The PID controller has gained popularity since 1942 due to Ziegler–Nichols tuning formula. This tuning formula is capable enough in providing the perfect starting solution and, at many times, is able to provide the best result among various conventional tuning formulae [80].

The PID (proportional–integral–derivative) controller is widely used in various fields, such as control engineering. PID controller is the controller parameters tuning process. In a PID controller, each mode (proportional, integral, and derivative mode) has a gain to be tuned, giving three variables involved in the tuning process as a result [57]. Table 5 shows the classification of the articles surveyed related to the area of control engineering.

Figure 10 shows a statistic of the three sub-areas that were treated in this summary; as can be seen, 50% of the articles are of various applications of PID control, then the electrical systems is 33%, and finally 17% are other types of applications, such as controlling a BLCD motor and a doubly fed induction generator (DFIG).

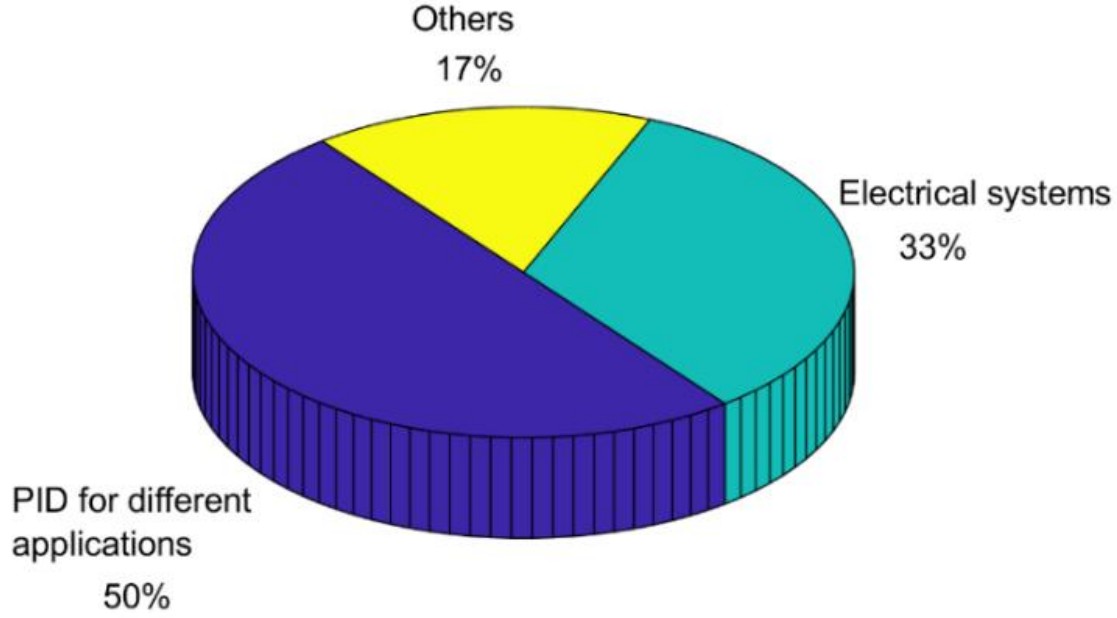

**Figure 10.** The graph represents the control subareas.

### 3.5. The TLBO Algorithm in Electromagnetism

In the area of electromagnetism there are only four articles that optimize with the TLBO algorithm. The electromagnetism is the relationship between the electric field and the magnetic field, in some of these articles the capacitor value and frequency radius were optimized using the TLBO algorithm and also the parameters to design the electrical machines. The TLBO algorithm guarantees the choice of the best solution to produce an optimal capacitor excitation to obtain the rated voltage at different loads, power factors, and speeds [69]. Table 6 shows the classification of the articles surveyed related to the area of electromagnetism.

**Table 5.** The TLBO algorithm used in control.

| Area | Ref | Year | Application | Description | Author's Conclusion |
|------|-----|------|-------------|-------------|---------------------|
| Control | [57] | 2014 | PID controller tuning for linear BLDC motor | TLBO aims to determine the parameters of the PID controller | TLBO manages to find the controller parameters efficiently |
| | [58] | 2018 | Electric vehicles | TLBO and PSO are used to optimize the integral controller gains and a comparison is made between the two-area hydrothermal unit with and without PEVs | The simulation shows that TLBO converges to the global minimum of the objective function |
| | [59] | 2015 | Fuzzy-PID controller for automatic generation control of a multi-area power system | TLBO aims to optimize the governor, turbine, load, and machine | Gives better performance, shorter settling time, less undershoot, less overshoot, and less frequency oscillation |
| | [60] | 2020 | Modelling and simulation to optimize direct power control of DFIGs in variable speed pumping power plants | TLBO aims to design and simulate a doubly fed induction generator | TLBO improves power quality by reducing stator active and reactive power ripple. THD of rotor and stator currents is lower |
| | [80] | 2020 | Design of optimal PID controller for varied systems using teaching–learning-based optimization | Design a PID controller to control the speed of a DC motor | The maximum percentage of over triggering is higher than with a typical PID controller; however, the rise and settling time is shorter |
| | [61] | 2015 | Charging frequency of multisource electrical systems | Optimize the gains of the PID controller and study its dynamic performance for the power system and compare it with the DE | It is more robust and stable to wide changes of system load, parameters, size, and location of load disturbance and various cost functions |
| | [62] | 2019 | Fractional-order PID controller optimized for the AGC (automatic generation control) of an interconnected power system | The thermal power system consists of a reheat turbine and power systems incorporated with GDBs and GRCs | The proposed FOPID controller using TLBO gives better results, compared to the PID controller |
| | [63] | 2015 | Control for an automatic voltage regulator | The objective function used is the integral absolute error multiplied by time (ITAE) and also ITSE | The controller parameters obtained are roughly similar and all show approximately similar Jmin values |
| | [64] | 2018 | Fractional order PID photovoltaic systems (FOPID) | All strategies are designed to control all cascading loops in the conversion chain in order to eliminate harmonics in the mains current | Simulations were performed to validate the functionality, robustness, and simplicity of the algorithm |
| | [65] | 2021 | Damping controller design of the STATCOM | TLBO aims to improve the dynamic stability of the power system under various operating conditions | The designed controller is robust and shows satisfactory performance to improve the dynamic stability of the power system |
| | [66] | 2014 | Auto-tuning control for a DVR compensator | TLBO aims to improve the THD and voltage drop rates of a sensitive load in the network | Simulation results show that TLBO is more efficient in its convergence speed and in the proposed optimal solution |
| | [67] | 2015 | Control of the generation of a power system using a 2DOF (2-degree freedom) PID controller | Demonstrate the advantages of TLBO over other techniques and the superiority of the 2DOF PID controller over the conventional PID | It is observed that the TLBO controller based on the 2DOF-PID controller is very effective and gives better performance, compared to others |
| | [68] | 2015 | Automatic control of the generation of multi-area electrical systems with various energy sources | Conduct a methodical simulation study to evaluate the performance of the proposed PID controller with the TLBO algorithm | The superiority of the proposed design approach has been demonstrated by comparing the results with some other techniques |

**Table 6.** The TLBO algorithm used in electromagnetism.

| Area | Ref | Year | Application | Description | Author's Conclusion |
|---|---|---|---|---|---|
| Electromagnetism | [69] | 2018 | The behavior of a self-excited induction generator | TLBO aims to minimize the error between load voltage and nominal value and compare the results with other algorithms | TLBO guarantees the choice of the best solution to produce an optimal capacitor excitation to obtain the rated voltage at different loads |
| | [70] | 2016 | Electromagnetic problems | The proposed technique is applied to two benchmarks related to the brushless DC wheel motor problem | The TLBO and ITLBO algorithms proved to be efficient in solving the problem with the advantage of not requiring control parameters |
| | [71] | 2019 | Sub-synchronous resonance elimination | An integral of time multiplied by the absolute value of the velocity deviation is taken as the objective function | When adequate additional transient power is supplied from DFIGs placed close to the power plants, the damping of torsional oscillations can be significant |
| | [72] | 2017 | Optimal design of electrical machines | Optimizing the equipment shop problem and switched reluctance motor (SRM) with flux barriers | In the solution of both problems, it was concluded that they are as efficient as other often-used algorithms |

### 3.6. The TLBO Algorithm in Digital Electronics

The subarea of digital electronics has three articles in one optimization algorithm based on evolutionary techniques that were considered for the optimal design perspective of the linear phase digital FIR filter for the better control of the filter parameters; in the other articles, they were looking to optimize the calibration of a camera and find the rotation coordinates and compare it with the efficiency with other algorithms, and, in the last one, it was the same to get a better control of the filter parameter. Table 7 shows the classification of the articles surveyed related to the area digital electronics.

**Table 7.** The TLBO algorithm used in digital electronics.

| Area | Ref | Year | Application | Description | Author's Conclusion |
|---|---|---|---|---|---|
| Digital Electronics | [73] | 2013 | LP and HP digital IIR filter design | TLBO aims to obtain the optimum values for the design of low-pass or high-pass filters | TLBO has a lower margin of error, compared to the other optimization methods |
| | [74] | 2020 | Camera calibration | Optimize the calibration of a camera and find the rotation coordinates and compare it with the efficiency with other algorithms | The TLBO algorithm was one of the four slowest algorithms in obtaining the values, however, it was one of the four that found the best results |
| | [75] | 2017 | Design of optimal FIR digital filters | Optimization algorithms were considered for the optimal design perspective of the linear phase digital FIR filter for better control of filter parameters | Jaya is better than the TLBO algorithm in terms of stopband attenuation and error values that can be observed in the simulation results |

*3.7. The TLBO Algorithm in Analog Electronics*

Design optimization of voltage-source inverters has been widely investigated in the literature; this section is about analogue electronics where a design of a triple-band antenna was designed; furthermore, in the area of an analogue filter, the algorithm can optimize values to obtain better behavior in different applications. Table 8 shows the classification of the articles surveyed related to the area of analog electronics.

**Table 8.** The TLBO algorithm used in analog electronics.

| Area | Ref | Year | Application | Description | Author's Conclusion |
|---|---|---|---|---|---|
| Analog Electronics | [76] | 2020 | Triple-band inverted F-antenna | TLBO aims to design an inverted F antenna that operates at certain specified frequencies | The antenna works satisfactorily in the EGSM-900, GSM-1800, and LTE-2600 frequency bands |
| | [77] | 2020 | Hybrid active power filter | A new clustering strategy is proposed to dynamically adjust the hierarchy of all individuals | This paper proposes a novel hierarchical TLBO (HTLBO) algorithm to accurately estimate the parameters of the hybrid active power filter |
| | [78] | 2016 | Optimal LC filter design | TLBO aims to minimize the total cost of the filter and heat sink | The measured TDD was 3.2% and the THDV was around 1%. The deviation of THDV value between calculation and measurement is mainly due to the effect of dead time |

## 4. Discussion

In the articles shown in all the tables, the efficiency of the TLBO algorithm in these type of applications can be observed, that is, in the calculation of the switching angles for the minimization of the THD percentage in the area of multilevel inverter; furthermore, in the other areas, it is a good option to optimize with. It is also observed that there are recent articles in all kinds of applications; this is due to the fact that the algorithm could be considered young or recent, which increases the interest of implementing and studying it in a wide variety of application areas. TLBO has been used to solve multiobjective optimization problems and has achieved some remarkable results. Therefore, studying and extending multiobjective variants of the TLBO algorithm to solve multiobjective problems is also a challenge for future researchers interested in this algorithm.

## 5. Conclusions

In this paper, an attempt has been made to provide an introduction and survey of the teaching–learning-based optimization (TLBO) algorithm. The TLBO algorithm is a metaheuristic method that allows the solving optimization of problems that could be considered a young algorithm, since it has 11 years of its creation. Reviewing these papers, we can see that the main work on TLBO has focused mainly on improving optimization performance and broadening application areas. Researchers have developed several variants of TLBO based on modifications and hybridizations to improve the optimization performance of TLBO; however, the original TLBO algorithm shows good behavior and has been successfully applied to several optimization areas. It is a practical and fast algorithm, since it does not require a long study of the behaviour of the algorithm when you change the parameters because one of the most important characteristics is that is parameterless; the only information that the user has to introduce are the population, design variables, maximum number of iterations, and the objective function.

We hope that this study will be useful to readers interested in the TLBO algorithm and its applications within the area of electronics.

**Author Contributions:** K.Y.G.D. Data curation, Formal analysis, Investigation, Writing; S.E.D.L.A. Conceptualization, Project administration, Supervision, Writing; J.A.A. Formal analysis, Methodology, Supervision, Validation; M.P.-S. Resources, Software, Visualization, Writing; V.H.O.P. Funding acquisition, Resources, Visualization, Software. All authors have read and agreed to the published version of the manuscript.

**Funding:** This research received no external funding.

**Institutional Review Board Statement:** Not applicable.

**Informed Consent Statement:** Not applicable.

**Data Availability Statement:** Not applicable.

**Conflicts of Interest:** The authors declare no conflict of interest.

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
