# Peer review of "Teaching–Learning-Based Optimization Algorithm Applied in Electronic Engineering: A Survey"

_electronics, doi:10.3390/electronics11213451_

Round 1
Reviewer 1 Report
The title of the reviewer paper looks interesting, but it poorly corresponds to the body of the paper. Now, the paper is not usable for engineers. It is dedicated for the library staff.
In my opinion, this paper could be accepted for publication after major revision.
In the revised version of this paper the Authors should take into account following remarks:
1. Fig. 2 is of poor quality and it should be improved.
2. At the end of line 122 is an extra text, which should be removed.
3. The aim of the paper should be clearly stated in Introduction.
4. In each review paper some information about cited papers should be presented. The Authors should add short information what specific is given in each of the cited papers.
5. The section Conclusion should be extended. It should be given some information about specific application of the considered algorithm.
Author Response
Reviewer 1
Electronics Sep28th, 2022
Dear Reviewer 1:
Subject: Submission of revised paper entitled "Teaching-Learning-Based Optimization Algorithm Applied in Electronic Engineering: A Survey". Submission no: 1935768.
Thank you for your valuable comments. We have carefully reviewed the comments and have revised the manuscript accordingly. Our responses are given in a point-by-point manner below. Changes to the manuscript are highlighted in yellow color.
Sincerely,
Dr. Susana Estefany De León Aldaco
Corresponding author
- Reviewer 1
Dear Reviewer
Thank you very much for considering that our article would at least be useful to a librarian. However, we would like to broaden the scope and usefulness of our article to a wider number of readers and for that reason we made several modifications in response to your comments.
Considering your valued comment. The authors have not made an exhaustive review because of the type of study according to some characteristics the one that is deeper and uses analysis techniques is the systematic review, according to that we follow the characteristics of a survey that collect relevant literature, give some comparison and conclusions about the weaknesses and strengths of a theory or method [1]. According to that we have change all the categories tables and conclusion as follows:
In the Tables II, IV, V, VI and VII:
The authors added two columns named as Description and Authors conclusions to show more about each article reviewed.
Conclusion in line 422 we added the next two paragraphs:
The analysis of the TLBO algorithm concludes that it is an efficient optimization method, capable of solving and optimizing single and multi-objective problems (with one or more objective functions), with various constraints or even without constraints. It is a practical algorithm since it does not require any adjustment of control parameters, which facilitates its implementation in various types of problems. However, it was also shown that it is a slow algorithm in comparison with others but capable of being implemented even in real time for the optimization of a problem.
We hope that this study will be useful for readers interested in the TLBO algorithm and its applications within the area of ​​electronics. Because the authors summarize the taxonomy in nine steps that can make it easy for readers to use this algorithm in different types of problems. And they can even search for the category to which their problem belongs and find if someone has already solved it and analyze how they did it, what points are pending for future research and from there work to generate greater contributions to the area.
[1] M. J. Foster and S. T. Jewell, Assembling the pieces of a systematic review: a guide for librarians. Rowman & Littlefield, 2017.
In my opinion, this paper could be accepted for publication after major revision.
Response: thank you for your comment, we have worked very hard to make changes to improve the article, we hope you like this improved version.
- 2 is of poor quality and it should be improved.
Response: Considering your comment. Figure 2 has been improved to make it easier to read, it is shown below.
Figure 1. Flowchart of the TLBO algorithm [1].
- At the end of line 122 is an extra text, which should be removed.
Response: Taking into account your comment. That paragraph was corrected as follows:
Results may be divided into subheadings. It should provide a concise and precise description of the experimental tests that the authors had made, their interpretation, as well as the experimental conclusions that can be drawn.
- The aim of the paper should be clearly stated in Introduction.
Response: In response to this commentary, the section of the Introduction was modified, a new paragraph was added to make clear the aim of the paper. They are as follows:
This paper focuses on providing a survey of recent optimization work and application of the TLBO optimization method in the area of electrical engineering, as opposed to previous reviews that exist in the literature that summarize all areas in general. The main contributions of this paper are the following points:
- Through the experience of the authors of this article in the search for abstracts of the TLBO algorithm it was noticed that it is difficult to find information collected on this algorithm, this is essential to become familiar with this method, for this reason, the objective of this article is to provide a study on some of the advances and applications that have been made with this algorithm in the field of electronic engineering.
- Some of the difficulties that may be encountered when aiming to optimize using a metaheuristic method is the taxonomy of the algorithms, which is why this article presents the essential steps to follow so that readers can apply and understand the TLBO algorithm.
- This article aims to benefit future research and practical applications by describing the various fields of application and the solutions obtained in the area of electronics.
- In each review paper some information about cited papers should be presented. The Authors should add short information what specific is given in each of the cited papers.
Response: Considering your comment. All the tables that summarized the information have 2 more columns where the authors added the description and conclusion that the writers have in their article, the tables are as follows:
TABLE II
the tlbo algorithm used in power electronics.
|
Area |
Ref |
Year |
Application |
Description |
Author’s conclusion |
|
Power Electronics |
[9] |
2020 |
DAB converter |
TLBO used to design the system with a minimum volume inverter and achieve maximum power efficiency |
TLBO has better results and a faster convergence rate |
|
[10] |
2019 |
PFC and cloud data logger monitoring |
TLBO used to optimize the capacitor bank and compensate power factor |
TLBO reduced the time to find the best solution and obtained the power factor near the desired level |
|
|
[11] |
2017 |
Load dispatch with TLBO |
TLBO used to reduce energy production costs in power plants through economical load dispatching |
TLBO provides lower-cost and high-quality solutions to the problem of economical freight forwarding |
|
|
[12] |
2015 |
Design of a hybrid energy system |
TLBO is used to design a hybrid system that can supply a typical household load. |
TLBO was able to find the optimal design parameters and the system can provide power at an acceptable cost |
|
|
[13] |
2016 |
Distribution network |
Optimize system performance, such as real power loss, voltage profile and voltage stability index |
TLBO has shown good performance and efficiency compared to the PSO and GA method |
|
|
[14] |
2016 |
A DSTATCOM |
Optimize the PI controller gains and filter parameters and compare the efficiency of the algorithms (TLBO and GEM) |
TLBO performs better and converges faster than the GEM. The current THD is below the IEEE 519 standard. |
|
|
[15] |
2016 |
Maximum power point global tracking (MPPT) |
Improve TLBO algorithm for use in PV modules, compare results with other algorithms |
I-TLBO allows to automatically adjust the teacher's values, but it takes longer than the TLBO, but it does increase its power by 0.6W |
|
|
[16] |
2019 |
Dynamic voltage recloser (DVR) design |
TLBO for the design to select the optimum nominal voltage values of the DVRs used in an IDVR to compensate for balanced voltage drops |
Both methods lead to the optimal solution. However, the accuracy of TLBO is higher than GA and in less time |
|
|
[17] |
2017 |
Flexible alternating current transmission systems "STATCOM" |
Minimize transmission losses, determine the optimum value of control variables (real and reactive power) |
TLBO has the efficiency to reduce the active power loss reasonably without violating any restrictions. It also has excellent convergence characteristics |
|
|
[18] |
2016 |
Maximum power point trackers (MPPT) |
The TLBO algorithm seeks to optimize the output power of the PV array as a function of the duty cycle of the DC-DC converter |
The TLBO algorithm converges slower than the MBA, so it is less efficient |
|
|
[19] |
2019 |
Parameter extraction from photovoltaic models |
Extract the parameters of photovoltaic models using ITLBO and compare their performance with TLBO and others |
ITLBO can provide more accurate and reliable parameter values, but the tables show that in some categories it has the same result as TLBO |
|
|
[20] |
2017 |
Modified energy-efficient localization |
Application of TLBO in MDV-Hop, Calculation and modification of the hop size, Optimal selection of anchor nodes, Updating the location |
TLBO achieves higher localization accuracy. The results show that it has better positioning coverage and high location accuracy with lower power consumption |
|
|
[21] |
2021 |
AC/DC hybrid power system |
The evaluation function examines the suitability of the degree point as a solution to the optimization problem |
It is tested on IEEE14, 30 and 57 bus systems and it is observed in the tables that TLBO is the reduction of TFC, TRPL and HVSI |
|
|
[22] |
2017 |
Operational scheduling of renewable energy sources |
TLBO aims to optimize the search for the lowest total cost of operation for 24 hours |
The results showed that TLBO is an effective method and achieves a lower cost (7-11%) than the other methods |
|
|
[23] |
2017 |
Operation of the pumped-storage hydroelectric power plant |
TLBO seeks to minimize the operating cost of the hydroelectric power plant |
TLBO obtains the optimal generation considering the wind farm and the PHS unit |
|
|
[24] |
2017 |
MPPT technology in the solar photovoltaic system |
TLBO aims to obtain at any time the maximum power from a photovoltaic panel and to design an efficient DC-DC converter |
The proposed TLBO-MPPT method gives more efficient results in the designed DC-DC converter compared to conventional techniques |
|
|
[25] |
2017 |
Hydrothermal energy system |
TLBO aims to determine the optimal power generation from hydro and thermal power plants to minimize the total operating cost of thermal power plants |
The results have been obtained for water discharge, reservoir storage volume and optimal MW values of real hydroelectric and thermal power |
|
|
[26] |
2016 |
Reconfiguration of the primary power distribution system |
1. Aims to develop a TLBO-based bit relocation controller for simultaneous reconfiguration of primary PDSs |
The best-reconfigured system has an active power loss of 0.130 MW, which is reduced by 30.10% with respect to the base network |
|
|
[27] |
2018 |
Real power generation for congestion |
TLBO aims to minimize congestion cost while satisfying the network constraint |
Total congestion cost ($/h) = 494.66 |
|
|
[28] |
2015 |
DC-DC converter on the RFID tag |
TLBO aims to increase the voltage conversion ratio and also reduce the size of the charge transfer capacitor |
The output voltage and voltage conversion ratio of the 4-stage DC-DC converter, are 7.741 V and 86.01% |
|
|
[29] |
2017 |
Electrical systems troubleshooting |
TLBO aims to estimate the location of faults in electrical systems and also to obtain the error rate |
TLBO achieves almost the same accuracy in a very short time which is 7 s and can detect and locate any type of fault |
|
|
[30] |
2015 |
Small-signal stability of a wind system with DFIG |
TLBO aims to formulate state space model of DFIG, investigation of fault passage capability |
Using TLBO is much better than PSO for the power system model |
|
|
[31] |
2016 |
Simulation of global MPPT for photovoltaic systems |
TLBO aims to track the global MPP (maximum power points) of PV power system under partial shading condition (PSC) |
Simulations performed under varying shading patterns reveal that the performance of the TLBO-based tracker is better than PSO and fuzzy logic control (FLC) |
|
|
[32] |
2016 |
Reactive power planning |
TLBO is applied to determine the size of reactive power sources |
From the results obtained, it is found that the active power loss is minimal in the TLBO method |
|
|
[33] |
2017 |
Microgrid energy management |
TLBO aims to develop an energy management infrastructure for sources and loads such that the total cost required is reduced |
TLBO can achieve very accurate results in virtually any circumstance. Compared to others, it is the best solution finder. |
TABLE IV
the tlbo algorithm used in Control.
|
Area |
Ref |
Year |
Application |
Description |
Author’s conclusion |
|
Control |
[50] |
2014 |
PID controller tuning for linear BLDC motor |
TLBO aims to determine the parameters of the PID controller |
TLBO manages to find the controller parameters efficiently |
|
[51] |
2018 |
Electric vehicles |
TLBO and PSO are used to optimize the integral controller gains and a comparison is made between the two-area hydrothermal unit with and without PEVs |
The simulation shows that TLBO converges to the global minimum of the objective function |
|
|
[52] |
2015 |
Fuzzy-PID controller for automatic generation control of a multi-area power system |
TLBO aims to optimize the governor, turbine, load and machine |
Gives better performance, shorter settling time, less undershoot, less overshoot and less frequency oscillation |
|
|
[53] |
2020 |
Modelling and simulation to optimize direct power control of DFIGs in variable speed pumping power plants |
TLBO aims to design and simulate a doubly-fed induction generator |
TLBO improves power quality by reducing stator active and reactive power ripple. THD of rotor and stator currents is lower |
|
|
[49] |
2020 |
Design of optimal PID controller for varied systems using teaching–learning-based optimization |
Design a PID controller to control the speed of a DC motor |
The maximum percentage of over triggering is higher than with a typical PID controller, however, the rise and settling time is shorter |
|
|
[54] |
2015 |
Charging frequency of multisource electrical systems |
Optimize the gains of the PID controller and study its dynamic performance for the power system and compare it with the DE |
It is more robust and stable to wide changes of system load, parameters, size and location of load disturbance and various cost functions |
|
|
[55] |
2019 |
Fractional-order PID controller optimized for the AGC (Automatic Generation Control) of an interconnected power system |
The thermal power system consists of a reheat turbine, and power systems incorporated with GDBs and GRCs |
The proposed FOPID controller using TLBO gives better results compared to the PID controller |
|
|
[56] |
2015 |
Control for an automatic voltage regulator |
The objective function used is the integral absolute error multiplied by time (ITAE) and also ITSE |
The controller parameters obtained are roughly similar and all show approximately similar Jmin value |
|
|
[57] |
2018 |
Fractional order PID photovoltaic systems (FOPID) |
All strategies are designed to control all cascading loops in the conversion chain in order to eliminate harmonics in the mains current |
Simulations were performed to validate the functionality, robustness and simplicity of the algorithm |
|
|
[58] |
2021 |
Damping controller design of the STATCOM |
TLBO aims to improve the dynamic stability of the power system under various operating conditions |
The designed controller is robust and shows satisfactory performance to improve the dynamic stability of the power system |
|
|
[59] |
2014 |
Auto-tuning control for a DVR compensator |
TLBO aims to improve the THD and voltage drop rates of a sensitive load in the network |
Simulation results show that TLBO is more efficient in its convergence speed and in the proposed optimal solution |
|
|
[60] |
2015 |
Control of the generation of a power system using a 2DOF (2-Degree Freedom) PID controller |
Demonstrate the advantages of TLBO over other techniques and the superiority of the 2DOF PID controller over the conventional PID |
It is observed that the TLBO controller based on the 2DOF-PID controller is very effective and gives better performance compared to other |
|
|
[61] |
2015 |
Automatic control of the generation of multi-area electrical systems with various energy sources |
Conduct a methodical simulation study to evaluate the performance of the proposed PID controller with the TLBO algorithm |
The superiority of the proposed design approach has been demonstrated by comparing the results with some other techniques |
TABLE V
the tlbo algorithm used in electromagnetism.
|
Area |
Ref |
Year |
Application |
Description |
Author’s conclusion |
|
Electromagnetism |
[62] |
2018 |
The behavior of a self-excited induction generator |
TLBO aims to minimize the error between load voltage and nominal value and compare the results with other algorithms |
TLBO guarantees the choice of the best solution to produce an optimal capacitor excitation to obtain the rated voltage at different loads |
|
[63] |
2016 |
Electromagnetic problems |
The proposed technique is applied to two benchmarks related to the brushless DC wheel motor problem |
The TLBO and ITLBO algorithms proved to be efficient in solving the problem with the advantage of not requiring control parameters |
|
|
[64] |
2019 |
Subsynchronous resonance elimination |
An integral of time multiplied by the absolute value of the velocity deviation is taken as the objective function |
When adequate additional transient power is supplied from DFIGs placed close to the power plants, the damping of torsional oscillations can be significantly |
|
|
[65] |
2017 |
Optimal design of electrical machines |
Optimizing the equipment shop problem and switched reluctance motor (SRM) with flux barriers |
In the solution of both problems it was concluded that they are as efficient as other often used algorithms |
TABLE VI
the tlbo algorithm used in digital electronics.
|
Area |
Ref |
Year |
Application |
Description |
Author’s conclusion |
|
Digital Electronics |
[66] |
2013 |
LP and HP digital IIR filter design |
TLBO aims to obtain the optimum values for the design of low-pass or high-pass filters |
TLBO has a lower margin of error compared to the other optimization methods |
|
[67] |
2020 |
Camera calibration |
Optimize the calibration of a camera and find the rotation coordinates and compare it with the efficiency with other algorithms |
The TLBO algorithm was one of the four slowest algorithms in obtaining the values, however it was one of the four that found the best results |
|
|
[68] |
2017 |
Design of optimal FIR digital filters |
The tlbo algorithm aims at obtaining the optimal design of the linear phase digital FIR filter for better control of filter parameters |
Jaya is better than the TLBO algorithm in terms of stopband attenuation and error values |
TABLE VII
the tlbo algorithm used in analog electronics.
|
Area |
Ref |
Year |
Application |
Description |
Author’s conclusion |
|
Analog Electronics |
[69] |
2020 |
Triple-band inverted F-antenna |
TLBO aims to design an inverted F antenna that operates at certain specified frequencies |
The antenna works satisfactorily in the EGSM-900, GSM-1800 and LTE-2600 frequency bands |
|
[70] |
2020 |
Hybrid active power filter |
A new clustering strategy is proposed to dynamically adjust the hierarchy of all individuals |
This paper proposes a hierarchical TLBO (HTLBO) algorithm to accurately estimate the parameters of the hybrid active power filter |
|
|
[71] |
2016 |
Optimal LC filter design |
TLBO aims to minimize the total cost of the filter and heat sink |
The measured TDD was 3.2% and the THDV was around 1%. The deviation of THDV value between calculation and measurement is mainly due to the effect of dead time |
- The section Conclusion should be extended. It should be given some information about specific application of the considered algorithm.
Response: In response to this observation, two paragraphs were added. The paragraphs are the following:
The analysis of the TLBO algorithm concludes that it is an efficient optimization method, capable of solving and optimizing single and multi-objective problems (with one or more objective functions), with various constraints or even without constraints. It is a practical algorithm since it does not require any adjustment of control parameters, which facilitates its implementation in various types of problems. However, it was also shown that it is a slow algorithm in comparison with others but capable of being implemented even in real time for the optimization of a problem.
We hope that this study will be useful for readers interested in the TLBO algorithm and its applications within the area of ​​electronics. Because the authors summarize the taxonomy in nine steps that can make it easy for readers to use this algorithm in different types of problems. And they can even search for the category to which their problem belongs and find if someone has already solved it and analyze how they did it, what points are pending for future research and from there work to generate greater contributions to the area.

Reviewer 2 Report
This paper presents a survey of the articles published in the period 2013-2021 related to applying the Teaching-Learning-Based Optimization (TLBO) which reproduces the dynamics that 14 occur in a classroom with the teacher and the student.Plenty papers were reviewed and some graphs were generated to summarize the most relevant of these articles. However, these papers should be furtherly summarized with definite conclusions. Besides, the motivation of this paper is unclear. The author mentioned “We hope that this study will be useful to readers interested in the TLBO algorithm and its applications within the area of electronics’, which however lacks support from prior work. The authors are recommended to elaborate on this part and clearly present the research gap.
Author Response
Reviewer 2
Electronics Sep28th, 2022
Dear Reviewer 2:
Subject: Submission of revised paper entitled "Teaching-Learning-Based Optimization Algorithm Applied in Electronic Engineering: A Survey". Submission no: 1935768.
Thank you for your valuable comments. We have carefully reviewed the comments and have revised the manuscript accordingly. Our responses are given in a point-by-point manner below. Changes to the manuscript are highlighted in yellow color.
Sincerely,
Dr. Susana Estefany De León Aldaco
Corresponding author
- Reviewer 2:
This paper presents a survey of the articles published in the period 2013-2021 related to applying the Teaching-Learning-Based Optimization (TLBO) which reproduces the dynamics that 14 occur in a classroom with the teacher and the student. Plenty of papers were reviewed and some graphs were generated to summarize the most relevant of these articles. However, these papers should be furtherly summarized with definite conclusions. Besides, the motivation of this paper is unclear. The author mentioned “We hope that this study will be useful to readers interested in the TLBO algorithm and its applications within the area of electronics, which however lacks support from prior work. The authors are recommended to elaborate on this part and clearly present the gap.
Response: We appreciate your valuable comments. And to solve all the comments the authors have rewritten and added some paragraphs to give more information about the aim of the work, give more information about the papers that were reviewed and the conclusion was extended as follows:
Introduction
This paper focuses on providing a survey of recent optimization work and application of the TLBO optimization method in the area of electrical engineering, as opposed to previous reviews that exist in the literature that summarize all areas in general. The main contributions of this paper are the following points:
- Through the experience of the authors of this article in the search for abstracts of the TLBO algorithm it was noticed that it is difficult to find information collected on this algorithm, this is essential to become familiar with this method, for this reason, the objective of this article is to provide a study on some of the advances and applications that have been made with this algorithm in the field of electronic engineering.
- Some of the difficulties that may be encountered when aiming to optimize using a metaheuristic method is the taxonomy of the algorithms, which is why this article presents the essential steps to follow so that readers can apply and understand the TLBO algorithm.
- This article aims to benefit future research and practical applications by describing the various fields of application and the solutions obtained in the area of electronics.
Conclusion
The analysis of the TLBO algorithm concludes that it is an efficient optimization method, capable of solving and optimizing single and multi-objective problems (with one or more objective functions), with various constraints or even without constraints. It is a practical algorithm since it does not require any adjustment of control parameters, which facilitates its implementation in various types of problems. However, it was also shown that it is a slow algorithm in comparison with others but capable of being implemented even in real time for the optimization of a problem.
We hope that this study will be useful for readers interested in the TLBO algorithm and its applications within the area of ​​electronics. Because the authors summarize the taxonomy in nine steps that can make it easy for readers to use this algorithm in different types of problems. And they can even search for the category to which their problem belongs and find if someone has already solved it and analyze how they did it, what points are pending for future research and from there work to generate greater contributions to the area.

Reviewer 3 Report
This paper presents a survey of the articles published in 2013-2021 related to applying Teaching-Learning-Based Optimization. The authors reviewed 62 papers and classified them into five categories. However, the authors did not analyse these papers. Therefore, I cannot advise that this paper is accepted for publication.
There are some comments on this manuscript:
In the abstract, the authors should present the key results of this study.
It is better to explain why the authors did this survey in the introduction.
There are five categories, not six. The authors should check it carefully.
The authors used the algorithm to optimize some objective functions related to the design in the electronics field. However, there is no description of this method or algorithm.
The authors should do the process of review article procedure.
The authors have only presented the information from the collected papers. There was no analysis of these publications. Therefore, the authors did not present the key findings of this survey.
Author Response
Reviewer 2
Electronics Sep28th, 2022
Dear Reviewer 2:
Subject: Submission of revised paper entitled "Teaching-Learning-Based Optimization Algorithm Applied in Electronic Engineering: A Survey". Submission no: 1935768.
Thank you for your valuable comments. We have carefully reviewed the comments and have revised the manuscript accordingly. Our responses are given in a point-by-point manner below. Changes to the manuscript are highlighted in yellow color.
Sincerely,
Dr. Susana Estefany De León Aldaco
Corresponding author
- Reviewer 3:
- In the abstract, the authors should present the key results of this study.
Response: Considering your comment. The authors change the abstract and with this we respond the third comment too that is that there were five categories, not six, it can be seen in the next paragraph:
This paper presents a survey of the articles published in the period 2013-2021 related to applying the Teaching-Learning-Based Optimization (TLBO) which reproduces the dynamics that occur in a classroom with the teacher and the student. This paper uses the algorithm to optimize some objective functions related to the design in the electronics field. A total of 62 papers were reviewed and some graphs were generated to summarize the most relevant of these articles. These have been classified into five categories based on the areas of electronic engineering like power electronics, control, electromagnetism, digital electronics and analogue electronics. This article is composed of two stages, the first is a summary of the information on electronics, in general, encompassing all its areas and the second focuses on the algorithm applied to multilevel inverters, for each stage graphs and tables are shown. The analysis of the TLBO algorithm concludes that it is an efficient optimization method, capable of solving and optimizing single and multi-objective problems, with several constraints or even without constraints. It is a practical algorithm since it does not require any adjustment of control parameters.
- It is better to explain why the authors did this survey in the introduction.
Response: In response to your comment. The authors have written a paragraph and some points that can make clear the aim of the work, as follows:
This paper focuses on providing a survey of recent optimization work and application of the TLBO optimization method in the area of electrical engineering, as opposed to previous reviews that exist in the literature that summarize all areas in general. The main contributions of this paper are the following points:
- Through the experience of the authors of this article in the search for abstracts of the TLBO algorithm it was noticed that it is difficult to find information collected on this algorithm, this is essential to become familiar with this method, for this reason, the objective of this article is to provide a study on some of the advances and applications that have been made with this algorithm in the field of electronic engineering.
- Some of the difficulties that may be encountered when aiming to optimize using a metaheuristic method is the taxonomy of the algorithms, which is why this article presents the essential steps to follow so that readers can apply and understand the TLBO algorithm.
- This article aims to benefit future research and practical applications by describing the various fields of application and the solutions obtained in the area of electronics.
Line 128 we change this paragraph from the abstract to the introduction:
Electronic engineering has been getting a lot of relevance in world technological development for example in electric vehicles or generating electricity from renewable energy sources to counteract the environmental impact that non-renewable sources generate. The TLBO algorithm has attracted the interest of a large number of researchers due to its efficiency, speed, and low initialization parameter requirements.
- There are five categories, not six. The authors should check it carefully.
Response: Taking into account your comment. In first comment has been shown how do the authors correct the information. Thanks for the correction.
Six has change to five
- The authors used the algorithm to optimize some objective functions related to the design in the electronics field. However, there is no description of this method or algorithm.
Response: Following your suggested comment. The authors added some information of the taxonomy of the algorithm, summarized in 9 steps as follows:
All the previously mentioned allows us to say that the TLBO algorithm can be summarized in the following steps to solve an optimization problem:
- Establish the fitness function or objective function.
- Set initialization parameters and variable limits
- A random population is generated. The population is expressed as follows:
[1]
- Teacher phase: The mean of each of the variables or individuals is calculated using the following equation.
[2]
- The teacher is established by selecting the best solution for that iteration:
[3]
- The grade obtained in each variable for each student and a new mean should be calculated. Then the difference between two means is calculated with the following equation:
[4]
The Teacher Factor (TF) can be considered as 1 or 2.
- The values are updated by adding the difference to the old solution, this is described in the following equation:
[5]
- Learner’s phase: In this second phase of the algorithm, knowledge is transmitted through the interaction that exists between the students. This interaction is described with the following equation:
[6]
- The process will end only if the maximum number of iterations is completed, otherwise the whole process has to be repeated.
- The authors should do the process of review article procedure.
Response: Taking into account your comment. The authors have added a new paragraph in line 110 to explain a little bit of the process of the reviewed articles.
The article review process was as follows, first, we aimed to find articles that performed optimization of the switching angles in multilevel inverters in journals like IEEExplore, MDPI, Springer Nature, ScienceDirect, and SciELO, among others, it was noted that it was an algorithm that had a few years since its creation and therefore the number of articles was not so large and the search was opened to all those articles that had a focus in the area of electronics and that used the TLBO algorithm or compared others with this one. The authors' attention was drawn to the fact that in most of the literature there were good results when implementing the TLBO algorithm and there were not many studies that synthesized the information and gave clear steps of the taxonomy to understand the algorithm more easily. From there, classification and analysis of the conclusions reached by the authors of each article were started in order to have a clear idea of whether or not a problem can be optimized using this method and if good results can be expected from this optimization. The articles that were discarded were those that did not pertain to electronics or the optimization problem was mostly related to another area and included little in the work on a minimal application of electronic engineering. In this case, since the algorithm was developed in 2011, no article was rejected because of its publication date, since it was considered that it was only a few years since its development compared to other algorithms such as PSO, DE or GA.
- The authors have only presented the information from the collected papers. There was no analysis of these publications. Therefore, the authors did not present the key findings of this survey.
Response: Considering your valued comment. The authors have not made an exhaustive review because of the type of study according to some characteristics the one that is deeper and uses analysis techniques is the systematic review , according to that we follow the characteristics of a survey that collect relevant literature, give some comparison and conclusions about the weaknesses and strengths of a theory or method [1]. According to that we have change all the categories tables and conclusion as follows:
In the Tables II, IV, V, VI and VII:
The authors added two columns named as Description and Authors conclusions to show more about each article reviewed.
Conclusion in line 422 we added the next two paragraphs:
The analysis of the TLBO algorithm concludes that it is an efficient optimization method, capable of solving and optimizing single and multi-objective problems (with one or more objective functions), with various constraints or even without constraints. It is a practical algorithm since it does not require any adjustment of control parameters, which facilitates its implementation in various types of problems. However, it was also shown that it is a slow algorithm in comparison with others but capable of being implemented even in real time for the optimization of a problem.
We hope that this study will be useful for readers interested in the TLBO algorithm and its applications within the area of ​​electronics. Because the authors summarize the taxonomy in nine steps that can make it easy for readers to use this algorithm in different types of problems. And they can even search for the category to which their problem belongs and find if someone has already solved it and analyze how they did it, what points are pending for future research and from there work to generate greater contributions to the area.
[1] M. J. Foster and S. T. Jewell, Assembling the pieces of a systematic review: a guide for librarians. Rowman & Littlefield, 2017.

Round 2
Reviewer 1 Report
The Authors properly addressed all my remarks.
In my opinion, this article can be published in its current form.
Author Response
Response: Thank you for approving the article for publication. I appreciate your valuable comments which helped to improve this work.
Reviewer 2 Report
No more comments.
Author Response
Response. I appreciate your valuable comments which helped to improve this work.
Reviewer 3 Report
I observe that the authors have responded to some of the previous topics. Unfortunately, I still find several challenging elements in this manuscript.
The survey has been modified and extended to some degree. However, I still find it lacks relevant literature, and some of the referred papers are insufficient as they are dated some time back.
This is a review paper. However, the authors did not do the process review article procedure.
It still is not clear why did the authors do this survey.
The analyses of the collected papers were not presented.
The key findings of this survey were not understandable.
Author Response
Reviewer 3
Electronics Oct 13th, 2022
Dear Reviewer 3:
Subject: Submission of revised paper entitled "Teaching-Learning-Based Optimization Algorithm Applied in Electronic Engineering: A Survey". Submission no: 1935768.
Thank you for your valuable comments. We have carefully reviewed the comments and have revised the manuscript accordingly. Our responses are given in a point-by-point manner below. Changes to the manuscript are highlighted in yellow color.
Sincerely,
Dr. Susana Estefany De León Aldaco
Corresponding author
- Reviewer 3:
I observe that the authors have responded to some of the previous topics. Unfortunately, I still find several challenging elements in this manuscript.
Response: Thank you very much for your valued comments. It is unfortunate that we did not cover all of the challenging elements with the previous responses, however, we hope that the responses provided in this document will better address the new comments.
- The survey has been modified and extended to some degree. However, I still find it lacks relevant literature, and some of the referred papers are insufficient as they are dated some time back.
Response: Considering your appreciated comment. Regarding the year of publication, the authors did not reject any paper according to the date because is a novel algorithm and since its creation, there are only been 11 years, in 2011 and 2012 the papers did not belong to electronic engineering, when you speak about other algorithms that have been created like 30 years before its necessary to limit the years because there are a lot of papers that had work with them, but in the case of TLBO, there are fewer publications because it is recent.
- This is a review paper. However, the authors did not do the process review article procedure.
Response: Thanks for your comment, it is really helpful for the authors to improve the paper. It is not very common to observe the process review in a survey but the authors decided to add the next flow chart to be clearer about the whole process of reviewing.
- Methodology
As a starting point, a search in various databases for surveys, systematic reviews, and state of the art reviews focused on the use of the TLBO algorithm in different areas was carried out. The articles found are classified in table I.
TABLE I
reviews of the TLBO algorithm.
|
Ref |
Year |
Article |
Area |
|
[7] |
2019 |
A survey on teaching–learning-based optimization algorithm: short journey from 2011 to 2017 |
Electrical engineering, data mining, optimization and other applications |
|
[8] |
2015 |
A Short Survey on Teaching Learning Based Optimization |
Optimization Method for Continuous Non-Linear Large Scale Problems Constrained and Unconstrained Real Parameter Optimization Problems Shape and Size Optimization of Truss Structures With Dynamic Frequency Constraints |
|
[9] |
2018 |
A survey of teaching–learning-based optimization |
Manufacturing and operation research, Mechanical and electrical engineering, Civil engineering, Electronics and control engineering, Pattern recognition and image processing, Other areas |
|
[10] |
2019 |
A Survey of Application and Classification on Teaching-Learning-Based Optimization Algorithm |
General: Shop Scheduling, power systems, truss structures, multi-objective optimal, two-sided assembly line, others |
|
[11] |
2015 |
An improved teaching-learning-based optimization: briefly survey |
Economic Load Dispatch Problems |
|
[12] |
2012 |
Population based meta-heuristic techniques for solving optimization problems: A selective survey |
Location of automatic voltage regulators in distributed systems, Integer Programming for generation maintenance scheduling in power systems, Data Clustering, Economic Load dispatch problema with incommensurable objectives |
|
[13] |
2019 |
A survey on new generation metaheuristic algorithms |
Data classification, quadratic assignment problems, design procedure, size and shape of structures |
|
[14] |
2016 |
Review of applications of TLBO algorithm and a tutorial for beginners to solve the unconstrained and constrained optimization problems |
Encompasses many areas |
|
[15] |
2017 |
Applications of TLBO algorithm on various manufacturing processes: A Review |
Machining processes: Ultrasonic,Electro chemical, Electrical discharge, Laser beam, Electron-beam, Water-jet, Abrasive-jet |
|
[16] |
2017 |
Review of the Teaching Learning Based Optimization Algorithm |
Term hydrothermal scheduling problema, dynamic economic dispatch, Flow shop and Job shop scheduling, and others |
As a result of this first classification shown in Table I, the idea of developing a survey article on the applications of the TLBO algorithm in the field of electronic engineering arose because it was identified that there was a lack of such articles.
This article presents a synthesis of some applications of the Teaching Learning Based Optimization (TLBO) algorithm to the area of electronic engineering, analyzing and classifying publications from the period 2013 - 2021.
The search for publications was initiated in various databases and search engines such as IEEExplore, Springer Nature, ScienceDirect, and SciELO, among others. The universe of publications analyzed and classified is 62 papers related to optimization using the TLBO algorithm in the area of electronic engineering.
The selection process of the articles to be reviewed is shown in the flow chart of the Figure 2, where you can understand the steps followed for the selection of the 62 articles, one of the first steps was the selection of the keywords, which were "TLBO electronic engineering", "TLBO Multilevel Inverter", "Electronic TLBO", "Electronic optimization TLBO".
Figure 1. Flow chart of the selection of the articles to review.
- It still is not clear why did the authors do this survey.
Response: Thanks for the comments, the main idea of the authors was to work with this algorithm to optimize the switching angles to get a low percentage of THD and at the same time respect the modulation index that the user wants to get, according to that the authors look in the literature and reviews to observe how other researchers have used it, but the authors' attention was drawn to the fact that there were no surveys focused directly on electronics. And all papers showed that the TLBO algorithm was able to solve a lot of optimization problems, so the authors decide to bring all the information in an easy way, summarizing it in a survey for researchers that wanted to work with it. So, the authors added a paragraph in line 112 in the section of Methodology, as it follows:
As a starting point, a search in various databases for surveys, systematic reviews, and state of the art reviews focused on the use of the TLBO algorithm in different areas was carried out. The articles found are classified in table I.
TABLE I
reviews of the TLBO algorithm.
|
Ref |
Year |
Article |
Area |
|
[7] |
2019 |
A survey on teaching–learning-based optimization algorithm: short journey from 2011 to 2017 |
Electrical engineering, data mining, optimization and other applications |
|
[8] |
2015 |
A Short Survey on Teaching Learning Based Optimization |
Optimization Method for Continuous Non-Linear Large Scale Problems Constrained and Unconstrained Real Parameter Optimization Problems Shape and Size Optimization of Truss Structures With Dynamic Frequency Constraints |
|
[9] |
2018 |
A survey of teaching–learning-based optimization |
Manufacturing and operation research, Mechanical and electrical engineering, Civil engineering, Electronics and control engineering, Pattern recognition and image processing, Other areas |
|
[10] |
2019 |
A Survey of Application and Classification on Teaching-Learning-Based Optimization Algorithm |
General: Shop Scheduling, power systems, truss structures, multi-objective optimal, two-sided assembly line, others |
|
[11] |
2015 |
An improved teaching-learning-based optimization: briefly survey |
Economic Load Dispatch Problems |
|
[12] |
2012 |
Population based meta-heuristic techniques for solving optimization problems: A selective survey |
Location of automatic voltage regulators in distributed systems, Integer Programming for generation maintenance scheduling in power systems, Data Clustering, Economic Load dispatch problema with incommensurable objectives |
|
[13] |
2019 |
A survey on new generation metaheuristic algorithms |
Data classification, quadratic assignment problems, design procedure, size and shape of structures |
|
[14] |
2016 |
Review of applications of TLBO algorithm and a tutorial for beginners to solve the unconstrained and constrained optimization problems |
Encompasses many areas |
|
[15] |
2017 |
Applications of TLBO algorithm on various manufacturing processes: A Review |
Machining processes: Ultrasonic,Electro chemical, Electrical discharge, Laser beam, Electron-beam, Water-jet, Abrasive-jet |
|
[16] |
2017 |
Review of the Teaching Learning Based Optimization Algorithm |
Term hydrothermal scheduling problema, dynamic economic dispatch, Flow shop and Job shop scheduling, and others |
As a result of this first classification shown in Table I, the idea of developing a survey article on the applications of the TLBO algorithm in the field of electronic engineering arose because it was identified that there was a lack of such articles.
- The analyses of the collected papers were not presented.
Response: Dear reviewer, we have tried to present a summary of the information found in our search through tables and graphs.
We would like to emphasise that our objectives with this survey are as stated in the last part of the Introduction section:
“This paper focuses on providing a survey of recent optimization works and applications of the TLBO optimization method in the area of electrical engineering, as opposed to previous reviews that exist in the literature that summarize all areas in general. The main contributions of this paper are as follows:
- The paper aims to provide a survey on the recent progress and application of the TLBO algorithm in the area of electronics. This is rarely found in previous works, allowing beginners to become familiar with the TLBO algorithm.
- The article provides a taxonomy of the TLBO algorithm, which is useful for readers to understand and apply the TLBO algorithm.
- The article describes the fields of application, and the solutions obtained by the TLBO algorithm. All these are useful for understanding the algorithm and are expected to benefit both practical applications and future research.”
Furthermore, we would like to emphasise that we adhere to the definition of "Survey" provided by the ACM, which reads as follows: According to the Association for Computing Machinery, a survey is “A paper that summarizes and organizes recent research results in a novel way that integrates and adds understanding to work in the field. A survey article assumes a general knowledge of the area; it emphasizes the classification of the existing literature, developing a perspective on the area, and evaluating trends”[7]. In other words, it develops a perspective of the area but does not go into depth and analyze each of the articles as is done in systematic reviews or reviews of the state of the art.
[7] A. f. C. Machinery. (2022, October 13). Information and guidelines for reviewers. Available: https://dl.acm.org/journal/trets/reviewers
- The key findings of this survey were not understandable.
Response: Thanks for the comments, to solve this comment it is necessary to emphasis that the authors are presenting a survey that have different characteristics than a systematic review or a state-of-the-art review, that is the reason because the authors did not have a depth analysis of each article, so, the authors added a paragraph in line 93, as it follows:
The objective of this paper is achieved by conducting a literature survey. According to the Association for Computing Machinery, a survey is “A paper that summarizes and organizes recent research results in a novel way that integrates and adds understanding to work in the field. A survey article assumes a general knowledge of the area; it emphasizes the classification of the existing literature, developing a perspective on the area, and evaluating trends”[7]. In other words, it develops a perspective of the area but does not go into depth and analyze each of the articles as is done in systematic reviews or reviews of the state of the art.
[7] A. f. C. Machinery. (2022, October 13). Information and guidelines for reviewers. Available: https://dl.acm.org/journal/trets/reviewers

Round 3
Reviewer 3 Report
The author has made some improvements to the original manuscript. I think it could be published after some polish and refinement.